

# Simphony: simulating large-scale, rhythmic data

Jordan M. Singer[1], Darwin Y. Fu[1] and Jacob J. Hughey[1,2]

[1] Department of Biomedical Informatics, Vanderbilt University Medical Center, Nashville, TN, United States of America
[2] Department of Biological Sciences, Vanderbilt University, Nashville, TN, United States of America

## ABSTRACT

Simulated data are invaluable for assessing a computational method's ability to distinguish signal from noise. Although many biological systems show rhythmicity, there is no general-purpose tool to simulate large-scale, rhythmic data. Here we present Simphony, an R package for simulating data from experiments in which the abundances of rhythmic and non-rhythmic features (e.g., genes) are measured at multiple time points in multiple conditions. Simphony has parameters for specifying experimental design and each feature's rhythmic properties (e.g., amplitude and phase). In addition, Simphony can sample measurements from Gaussian and negative binomial distributions, the latter of which approximates read counts from RNA-seq data. We show an example of using Simphony to evaluate the accuracy of rhythm detection. Our results suggest that Simphony will aid experimental design and computational method development. Simphony is thoroughly documented and freely available at https://github.com/hugheylab/simphony.

# INTRODUCTION

Rhythms are ubiquitous across domains of life and across timescales, from hourly division of bacteria (*Cooper & Helmstetter, 1968*) to seasonal growth of trees (*Kramer, 1936*). These biological rhythms are often driven by systems of genes and proteins. Prominent examples are the systems underlying circadian rhythms, which have a period of approximately 24 h and have been observed in species across the biosphere (*Young & Kay, 2001*) and throughout the body of multicellular organisms (*Yoo et al., 2004*; *Zhang et al., 2014*).

To interrogate these rhythmic biological systems, researchers are increasingly using technologies that measure the abundance of thousands of molecules in parallel (e.g., quantifying the transcriptome by RNA-Seq). The critical decisions then become how to design the experiments and how to analyze the data. For example, there are now numerous methods for detecting rhythms in high-dimensional data (*Yang & Su, 2010*; *Hughes, Hogenesch & Kornacker, 2010*; *Thaben & Westermark, 2014*; *Wu et al., 2016*). A valuable aid to such decisions is simulation. In simulated data, unlike in experimental data, the ground truth is known (e.g., whether a gene is rhythmic). Consequently, to the extent that simulated data possess the essential features of experimental data, simulation can be used to estimate a method's ability to distinguish signal from noise (*Deckard et al., 2013*).

Corresponding author
Jacob J. Hughey,
jakejhughey@gmail.com

Simulated data are also typically faster and less expensive to generate than experimental data, especially omics data from high-resolution time courses.

Unfortunately, there is a shortage of publicly available tools for simulating rhythmic data. This forces researchers to create their own simulations from scratch (*Deckard et al., 2013*; *Singer & Hughey, 2018*) or to forgo simulations altogether. Although several tools exist to simulate particular types of transcriptome data (*Dembélé, 2013*; *Frazee et al., 2015*; *Zappia, Phipson & Oshlack, 2017*), most are not designed to simulate data from time-course experiments. One exception is Polyester (*Frazee et al., 2015*), which can simulate RNA-seq reads from multiple time points and conditions. However, Polyester models many aspects of the sequencing process, which incurs a computational burden and may not be directly relevant for designing experiments to collect rhythmic data or evaluating methods to analyze such data. Recognizing a gap, *Hughes et al. (2017)* recently developed CircaInSilico, a web-based application for simulating rhythmic data. Although CircaInSilico has a convenient user interface, it has several limitations—for example, the simulated rhythms can only be sinusoidal. In addition, even though read counts from RNA-seq data are often modeled using a negative binomial distribution (*Robinson & Smyth, 2007*), CircaInSilico can only simulate Gaussian noise. Thus, there is still a need for a flexible tool to simulate large-scale, rhythmic data.

To address this need, we developed a simulation package called Simphony. Simphony has adjustable parameters for specifying experimental design and modeling rhythms, including the ability to sample from Gaussian and negative binomial distributions. Simphony is implemented in R, thoroughly documented, and freely available at https://github.com/hugheylab/simphony .

## MATERIALS AND METHODS

### Simulating rhythmic data using Simphony

Simphony simulates experiments in which the abundances of rhythmic and non-rhythmic features (e.g., genes) are measured at multiple time points in one or more conditions (Table 1). Within a given simulated experiment (i.e., a simulation), the expected abundance $m$ of feature $i$ in condition $k$ at time $t$ is modeled as

$$m_{ik}(t) = a_{ik}(t) \cdot f_{ik}(\frac{2\pi}{\tau_{ik}} \cdot (t + \phi_{ik})) + b_{ik}(t),$$

where $a$ is the amplitude, $f$ is a periodic function with period $2\pi$ (by default, $f(\theta) = sin(\theta)$), $\tau$ is the period of rhythmic changes in abundance (by default, 24), $\phi$ is the phase, and $b$ is the baseline abundance. If $a$ and $b$ are constant and $f(\theta) = sin(\theta)$, the model is equivalent to cosinor. Time-dependent $a$ can create damped rhythms, whereas time-dependent $b$ can create drift. Non-rhythmicity is defined by $a = 0$.

Given $m_{ik}(t)$, Simphony samples measurements from one of two families of distributions: Gaussian and negative binomial. The former represents an idealized experimental scenario, whereas the latter approximates read counts from RNA-seq. For Gaussian sampling, the abundance of feature $i$ in sample $j$ belonging to condition $k$ follows

$$Y_{ij} \sim N(m_{ik}(t_j), \sigma_{ik}^2),$$
**Table 1  Available options in Simphony.**

| Type of parameter | Parameters |
|---|---|
| Experimental design | Time points: |
| | • First and last time points, interval, and number of samples per time point |
| | • Specified time points and number of samples per time point |
| | • Drawn from a uniform distribution, first and last possible time points, and total number of samples |
| | Number of conditions |
| | Number of features in each group |
| Properties of abundance (per feature group per condition) | Rhythmic shape |
| | Period |
| | Phase |
| | Amplitude (can be time-dependent) |
| | Baseline (can be time-dependent) |
| Sampling measurements | Family: |
| | • Gaussian |
| | • Negative binomial |
| | Standard deviation (if Gaussian) |
| | Mean-dispersion relationship (if negative binomial) |

where $\sigma^2$ is the variance (by default, 1). For negative binomial sampling, we follow a similar strategy to DESeq2 (*Love, Huber & Anders, 2014*) and Polyester (*Frazee et al., 2015*), such that

$$Y_{ij} \sim NB(\mu = 2^{m_{ik}(t_j)}, \alpha = g_{ik}(2^{m_{ik}(t_j)})),$$

where $\mu$ is the expected counts, $\alpha$ is the dispersion (the variance of a negative binomial distribution is $Var(Y) = \mu + \alpha\mu^2$), and $g$ is a function that maps expected counts to dispersion. The default $g$ was estimated from RNA-seq data from mouse liver (see the next section for details).

Experimental design in Simphony is specified in one of three ways: (1) first and last time points, interval between time points, and number of samples per time point per condition, (2) exact time points and number of samples per time point per condition, or (3) time points sampled from a uniform distribution, range of possible time points, and total number of samples per condition. By default, Simphony uses option (1), with first and last time points of 0 and 48, interval between time points of 3, and number of samples per time point of 2.

The Simphony R package has two dependencies: data.table (*Dowle & Srinivasan, 2018*) and foreach (*Calaway, Microsoft & Weston, 2017*).

## Estimating statistical properties of experimental RNA-seq data

To estimate the relationship between expected counts and dispersion in real RNA-seq data, we used PRJNA297287 (*Atger et al., 2015*). We used the samples that were collected in quadruplicate from livers of wild-type, ad libitum-fed mice every 2 h for 24 h in LD

12:12 (48 samples total). We downloaded the raw reads, then quantified gene-level counts using Salmon v0.11.3 (*Patro et al., 2017*) and tximport v1.8.0 (*Soneson, Love & Robinson, 2015*). We kept the 15,069 genes that had at least 10 counts in half of the samples. We used DESeq2 v1.20.0 to estimate parametric and local regression-based mean-dispersion curves (*Love, Huber & Anders, 2014*) (Fig. S1A). The input to DESeq2 included a design matrix based on cosinor regression, so that dispersion estimates were not biased by variation in expression due to a daily rhythm. Compared to the parametric mean-dispersion curve, the local regression-based curve had a considerably lower root-mean-squared error (0.94 compared to 1.09, in units of log dispersion), so we set it as the default in Simphony (*g* in the equation above). DESeq2 also provided an estimate of the variance of the residual log dispersion (around the curve). Finally, we used fitdistrplus v1.0-14 (*Delignette-Muller & Dutang, 2015*) to approximate the distribution of mean normalized counts as log-normal. The Simphony documentation includes an example of how to sample from the estimated distributions of residual log dispersion and mean normalized counts (Fig. S1B).

### Validating statistical properties of simulated data

We performed multiple simulations to validate the statistical properties of data generated by Simphony. Each simulation had time points spaced 0.1 h apart (period of 24 h), 100 samples per time point, and one feature for each combination of parameter values related to measurements. Simulations based on negative binomial sampling used the default function for calculating dispersion.

To validate mean and standard deviation, we simulated non-rhythmic abundance (amplitude of 0) based on Gaussian and negative binomial sampling. For the simulation using Gaussian sampling, we varied the desired mean and standard deviation. For the simulation using negative binomial sampling, we varied the desired mean $\log_2$ counts. In both cases, we then calculated the empirical mean and standard deviation (Table S1).

To validate amplitude and phase, we simulated rhythmic abundance based on Gaussian and negative binomial sampling (using the default $f(\theta) = sin(\theta)$). For both types of sampling, we varied the desired amplitude and phase. For the simulation based on Gaussian sampling, we used the limma R package v3.38.3 (*Smyth, 2004*; *Ritchie et al., 2015*) to fit each feature's abundance to a linear model that had terms for $cos\left(\frac{2\pi}{\tau}t\right)$ and $sin\left(\frac{2\pi}{\tau}t\right)$ (cosinor regression). We then used the model coefficients to estimate each feature's amplitude and phase according to the trigonometric identity $a \cdot cos\theta + b \cdot sin\theta = c \cdot sin(\theta + \phi)$, where $c = \sqrt{a^2 + b^2}$ and $\phi = \frac{\pi}{2} - atan2(b, a)$ (Table S2). For the simulation based on negative binomial sampling, we followed a similar procedure, except we log-transformed the counts before passing them to limma.

### Detecting rhythmicity in simulated data

We calculated gene-wise p-values of rhythmicity using JTK_CYCLE v3.1 (*Hughes, Hogenesch & Kornacker, 2010*) after transforming the expression values, sampled from the negative binomial family, using $\log_2(\text{counts}+1)$. We used the p-values and the precrec R package v0.9.1 (*Saito & Rehmsmeier, 2017*) to calculate the area under the receiver

operating characteristic (ROC) curve for distinguishing non-rhythmic genes from each group of rhythmic genes (specified by rhythm amplitude and baseline in $\log_2$ counts).

## RESULTS

To validate the statistical properties of data generated by Simphony, we simulated data covering a range of parameter values for the Gaussian and negative binomial families. To ensure that the properties approached their asymptotic values, time points were spaced 0.1 h apart (period of 24 h), each with 100 samples. For non-rhythmic abundance, we verified that the observed mean and standard deviation corresponded to the expected values (Table S1). For rhythmic abundance, we verified that the observed amplitude and phase corresponded to the expected values (Table S2).

To highlight Simphony's flexibility, we simulated gene expression from a variety of patterns, including ones in which the rhythm amplitude or baseline expression was time-dependent (i.e., non-stationary). For each pattern, we sampled expression values from the Gaussian and negative binomial families (Fig. 1). These patterns are only examples—Simphony can simulate data from any rhythmic waveform or non-stationary trend provided as a function in R. We also simulated an experiment in which each of 200 genes had a different rhythm amplitude and phase (Fig. 2), and an experiment having two conditions, in which genes' rhythms had a different amplitude, phase, or period in each condition (Fig. S2).

To show an example of Simphony's utility, we created simulations to quantify how the accuracy of rhythm detection depends on experimental and biological parameters. We simulated experiments having various intervals between time points and one sample per time point. Each simulation included 20,000 genes spanning a range of values for baseline expression and rhythm amplitude (including amplitude 0 for non-rhythmic genes) (Fig. S3A). Because Simphony is not designed to detect rhythmicity, we calculated each gene's $p$-value of rhythmicity in each simulation using JTK_CYCLE, then calculated the area under the ROC curve for distinguishing non-rhythmic genes from each group of rhythmic genes. As expected, rhythm detection improved as rhythm amplitude increased or the interval between time points decreased (Fig. 3A). Rhythm detection also improved as baseline expression increased (and thus as the standard deviation of log-transformed counts of non-rhythmic genes decreased; Fig. 3B and Fig. S3B).

## DISCUSSION

Simphony is a versatile framework for simulating rhythmic data. Although Simphony is especially apt for simulating transcriptome data, it is general enough to simulate data of various types (e.g., bioluminescence). A future objective is to use simulated data from Simphony to comprehensively benchmark computational methods for detecting rhythmicity. Simphony's flexibility will be key to mimicking the diversity of rhythms seen in practice. Simphony's ability to simulate non-stationary trends in particular is critical, since the possibility of non-stationarity is one reason the guidelines for genome-scale analysis of biological rhythms recommend collecting data from at least two cycles (*Hughes*

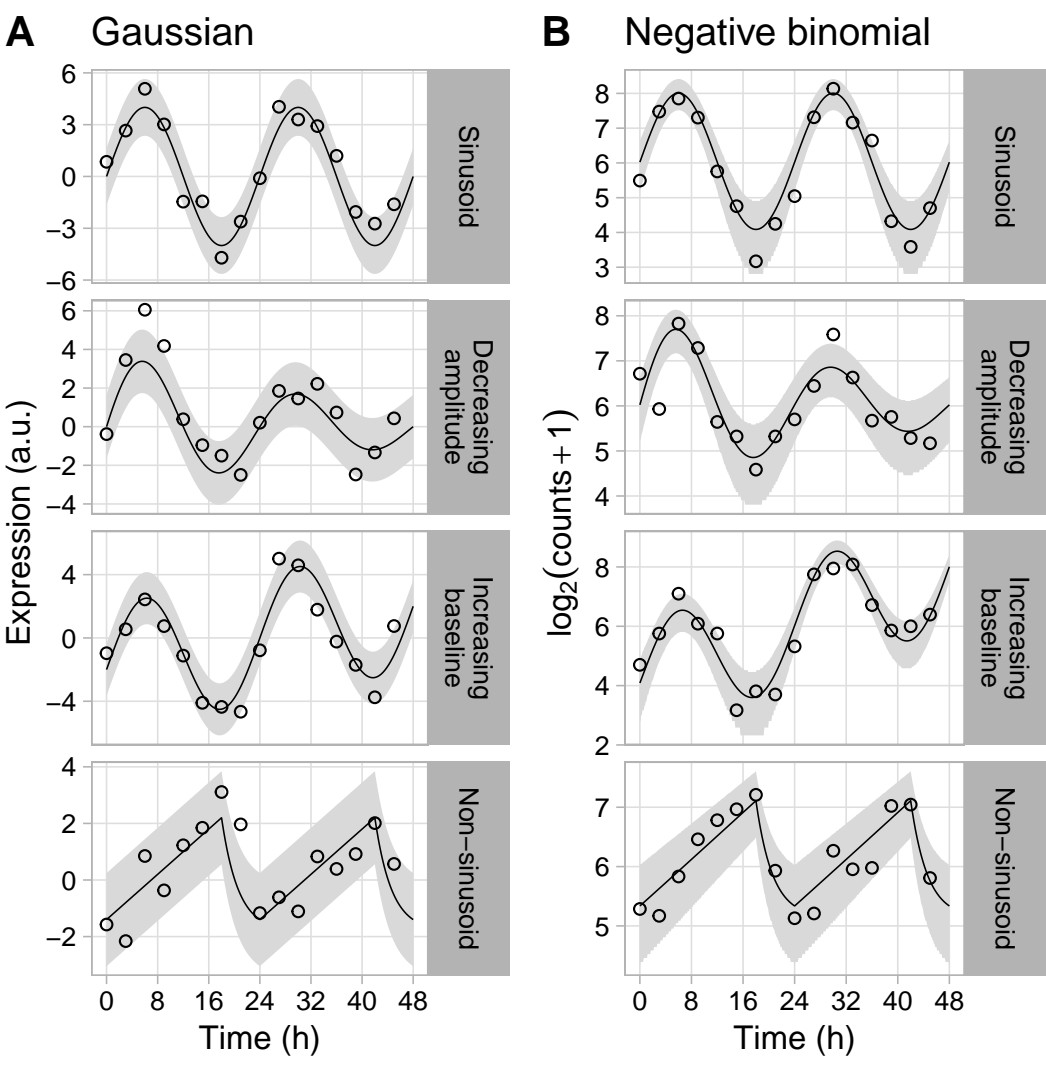

**Figure 1** **Examples of rhythmic data generated by Simphony.** Gene expression values (i.e., abundances) were sampled from the (A) Gaussian or (B) negative binomial family. Time points were spaced 3 h apart, with 1 sample per time point. Circles show the sampled gene expression values, black lines show the expected expression over time, and gray ribbons show the corresponding 90% prediction intervals. The prediction intervals for negative binomial sampling have discontinuities because the sampled values can only be integers greater than or equal to zero. The prediction intervals for negative binomial sampling also shrink as expected expression increases, due to the mean-dispersion relationship.

*et al., 2017*). Ultimately, we anticipate that Simphony will inform the design of experiments for interrogating rhythmic biological systems and the development of methods for analyzing data containing rhythmic signals.

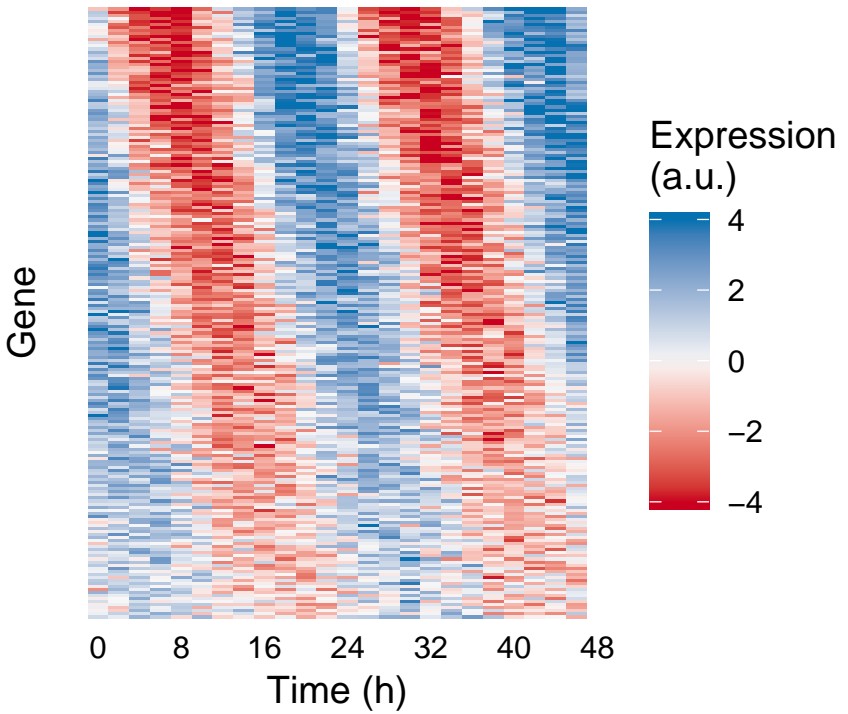

**Figure 2** **Example of medium-scale simulation in Simphony.** Expression values of 200 rhythmic genes, each gene with its own amplitude and phase, were sampled from the Gaussian family. Rhythms followed a sinusoid. Each row in the heatmap corresponds to a gene, each column to a time point. For ease of visualization, sampled expression values were cropped to be between −4 and 4.

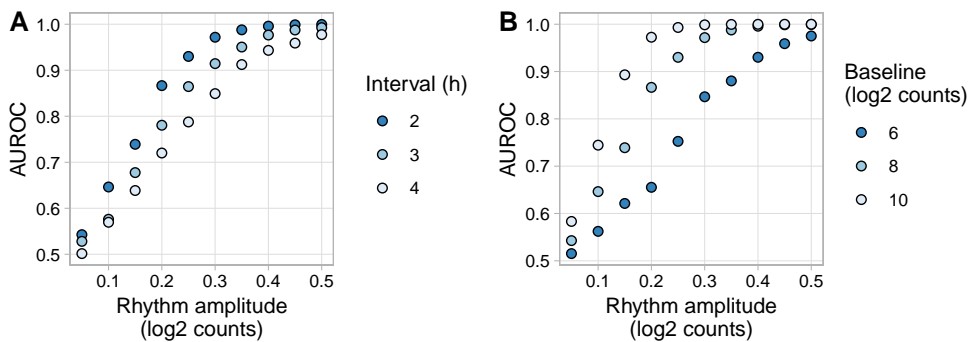

**Figure 3** **Example of evaluating rhythm detection using data generated by Simphony.** Simulations had various values of the interval between time points and one replicate per time point. Each simulation included 20,000 genes having various values of baseline expression and rhythm amplitude (including amplitude 0). Rhythms followed a sinusoid of period 24 h. Expression values were sampled from the negative binomial family. Gene-wise p-values of rhythmicity from JTK_CYCLE were used to calculate the area under the ROC curve (AUROC) for distinguishing non-rhythmic genes from each group of rhythmic genes. (A) AUROC vs. rhythm amplitude and interval, for genes with a baseline log2 counts of 8. (B) AUROC vs. rhythm amplitude and baseline expression, for the simulation with an interval of 2 h. AUROC of 0.5 corresponds to random detection, while AUROC of 1 corresponds to perfect detection.

### Funding

This work was supported by NIH R35GM124685 to Jacob J. Hughey. The funders had no role in study design, data collection and analysis, decision to publish, or preparation of the manuscript.

### Grant Disclosures

The following grant information was disclosed by the authors:
NIH: R35GM124685.

### Competing Interests

The authors declare there are no competing interests.

### Author Contributions

- Jordan M. Singer, Darwin Y. Fu and Jacob J. Hughey conceived and designed the experiments, performed the experiments, analyzed the data, prepared figures and/or tables, authored or reviewed drafts of the paper, approved the final draft.

### Data Availability

The simphony R package is available on GitHub: https://github.com/hugheylab/simphony. Data, code, and results for this study are available on Figshare: https://doi.org/10.6084/m9.figshare.7441355.

### Supplemental Information

Supplemental information for this article can be found online at http://dx.doi.org/10.7717/peerj.6985#supplemental-information.

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
