# Peer review of "Simphony: simulating large-scale, rhythmic data"

_PeerJ, doi:10.7717/peerj.6985_

## Round 0.1 · original submission · Major Revisions

The reviewers have made some positive comments about your manuscript, and there is clearly a need for new packages in this area, but they have also raised a number of issues, some of which I think you could address without too much difficulty and others which might require appreciable work/extensions to the package. If you feel that you can address each of the reviewers’ comments, I would be very happy to see a revised version of the manuscript.

Reviewer 1 ·

Basic reporting

A freely available R package for generating realistic circadian time series is certainly a welcome addition, to assist in testing new rhythmicity and circadian parameter estimation methods. The manuscript is clearly written.

Experimental design

There is a serious flaw in the design in that the length of the simulations is only 24h, one cycle, in length. Rhythmicity testing requires 2 cycles at the very least.

All the time series depicted in the main manuscript and in the supplement are all only 24 hours in length (the vignettes use only 24h as well). How can rhythmicity be tested with only one cycle? Just because a time series goes up and down over a day doesn’t mean it will do it again the next day. I’ve had experimental results where the first day looked like the first cycle of a rhythm, but then the next day was quite different, showing it was a transitory event and not a true rhythm (with a similar pattern across all samples, so multiple samples per time point doesn’t alleviate the problem). All examples should show at least 2 days.

See https://www.ncbi.nlm.nih.gov/pubmed/29098954 for guidelines and best practices, which at least one of the manuscript authors signed onto and is referenced in the text. This article states, "By definition, biological rhythms repeat. We therefore recommend collecting at least 2 complete cycles of data when detecting rhythmicity (i.e., 48 h for collections under constant conditions). The guiding principle behind this recommendation is that when identifying a rhythmic process, one would like to observe both the peak and trough repeat at least once. Simulations show that collecting fewer than 2 cycles in a time series makes the resulting data sensitive to outliers and can dramatically increase the number of false-negatives (see the “Synthetic Data for Benchmarking” section)."

For purposes of phase, waveform, and amplitude estimation, it's true that under LD entrained conditions 24h of data can be used. But rhythmicity cannot be correctly tested with only 1 day of data. How is limma detecting rhythmicity with only 1 day of data? Is it really testing for a sinusoidal waveform?

In addition, in trying out the package, I couldn’t find a way to simulate time series longer than 24h. *It’s really important that there be an easy option for specifying length of the time series simulations.* Note that CircaInSilico does provide this feature. For use in studies after constant conditions or for LD conditions in which entrainment may not be stable (short or long T-cycles, skeleton PP, jet lag experiments, etc), multiple cycles will be very important.

Validity of the findings

Due to the serious flaw of only one cycle being simulated, the rhythmicity results shown in Figure 2 are not valid.

Additional comments

For clarity, please state explicitly in lines 94-95 that the count-dispersion relation is being used to generate the function g_ik; this direct link will help the reader make the connection.

What choices does the user have: # days, time step, noise type and parameters, signal amplitude, phase, baseline, waveform, others? Perhaps include a table listing the options available, as a convenient summary showing the flexibility of the package.

FigS2: just an example to illustrate options? Two simulated series for each condition and gene? Can it accurately detect the amplitudes and phases? The purported goal of the package is to provide large simulated datasets to guide experimental design, so do some examples that mimic large-scale experiments. For instance, demonstrate how to use the simulated data sets to determine how many samples are needed per time point given a certain sampling rate like every 4 hours over 2 days to achieve a desired false discovery rate.

The current text seems a bit thin. Demonstrating its use by applying metacycle to a large simulated data set generated by symphony would nice to see, rather than only limma for the single validation example. Using a package like metacycle will provide an alternative rhythmicity test(s) as well as phase and amplitude estimates and could further expand the scenario in Figure S2 into an interesting test case (like determining what sampling rate with 2 days of data would be required for reliable estimation of phase).

For installing simphony, note that you can also simply type drat::addRepo('hugheylab')
Into the Console window. I couldn’t get the .Rprofile approach to work, but simply pasting that line into the Console worked fine.

Reviewer 2 ·

Basic reporting

Authors present an R package, called Simphony, to simulate large-scale circadian transcriptome data regarding different rhythmic features measured at multiple time points in multiple conditions. To do so, a measure of abundance is generated regarding a signal-noise model using two distribution families: the Gaussian (idealized experimental scenario) and Negative Binomial (which approximates read counts from RNA-Seq). A github repository contains all package dependencies and data.

This paper uses a suitable language and the structure and goals are clearly exposed. However, some statements require additional references and other times they are too old, see the comments made on lines 24,28 of the .pdf file. Literature review presents several gaps, see comments on the lines 34, 44 of the .pdf file.

The scientific question proposed in this paper, i.e. the simulation of circadian rhythmic genes imitating real data bases is relevant for biologists. However, this question has been already dealt in literature (see the comments on line 34 in the .pdf file); the novelty of the simulation model-based is not clear (see comments on line 60 in the .pdf file); and the procedure is only validated for symmetric shapes, despite that circadian data bases also include non-sinusoidal patterns (see line 143 in the .pdf file). Please, see Experimental Design for details.

Experimental design

As I mentioned above, this paper presents a meaningful research question and methods are described with sufficient details.
However, it presents several controversial points:

A.- This question has been previously studied in literature, both for microarray (Dembelé (2013) among others) and RNA-seq technologies (Zappia, Phipson and Oshlack (2017) among others). Authors do not refer/compare to this fact in the paper.

B.- The model employed to generate synthetic data is very close to Cosinor model, at least for the default case considered in this paper, i.e. when f is assumed to be a sinusoidal function. Authors should discuss the similarities/differences with this model. In addition, only this default (sinusoidal) function is considered to validate the procedure. It is not enough to replicate the wide variety of patterns that appear in practice.

C.- As expected, results show a suitable performance of the simulation tool. Note that data were generated from a sinusoidal model and the fittings are made with Cosinor model (based on sinusoidal shape too). However, non-sinusoidal (asymmetric) rhythms are found in circadian data bases (Thaben 2014). Although, authors consider a sawthooth wave, it is again just a symmetric pattern. More arbitrary and extreme patterns are required to validate the procedure.

Please, revise the attached .pdf file for specific details, comments and suggestions

Validity of the findings

The originality of the research question is questionable.
Conclusions depends on the periodic simulation function (which is always symmetric) and on the Cosinor regression model which intrinsically assumes sinusoidal shapes. Thus, those conclusion cannot be extended to circadian data bases where there are many arbitrary rhythmic patterns, not only sinusoidal.

Additional comments

Please, find comments, details and suggestions in the attached .pdf file

Annotated reviews are not available for download in order to protect the identity of reviewers who chose to remain anonymous.

Reviewer 3 ·

Basic reporting

.

Experimental design

.

Validity of the findings

.

Additional comments

Manuscript Title: Simphony: simulating large-scale, rhythmic data

The increasing number of large-scale experimental datasets available for time-course analysis requires computational methods that can distinguish true periodic signal from noise. Simulated data, where truth is known, help to evaluate how well a particular method performs. In response to the lack of freely available (and flexible) tools for simulating periodic data, the authors developed and benchmarked a new framework called Simphony.

Simphony represents a nice addition to the toolkit of any chronobiologist who designs or analyzes large-scale time-course experiments. Advantages over other publicly available tools are simulations that can more closely approximate experimental: 1) feature waveforms (not always sinusoid) and 2) distribution of RNAseq data (not always gaussian). As usual, the technical details of the work are superb. However, there are a some improvements that could improve the impact of the work [see below]. Although more flexible than current publicly available tools (i.e. CircaInSilico, JBR 2017), it is considerably less user-friendly. And, in terms of performance, it appears to be less flexible than prior published approaches. Improvements to either of these aspects would considerably increase the impact of Simphony as a resource and the paper as a consequence of that.

Major Comments

Different methods for rhythm detection usually perform best when validated against their own simulated datasets. There are few published papers that evaluate multiple methods using the same simulated dataset. In that sense, there is a need for a simulation framework that is both easy to use and generalizable, as the authors point out. Our concern, however, is that Simphony in its present form falls a bit short of meeting these needs.

● Simphony generates two rhythmic waveforms. Are more necessary? In 2013, Deckard A. et al. reported a simulation generating six different periodic patterns and two non-periodic patterns.
● While free to download and use in R stats, Simphony is still not as user-friendly as CircaInSilico (Hughes et al, JBR 2017), which can be run as a web-based App (no coding required). This could ease the learning curve and encourage wider adoption.
● The manuscript lacks explicit description of how Simphony could be used to help guide the design of a typical time-course experiment. The ROC curves (Fig 2) demonstrate nicely how cosinor regression better discriminates rhythmic from non-rhythmic features when the sampling interval is smaller (and amplitudes are higher). This, however, is pretty intuitive, and a conclusion reached in prior work (e.g. Deckard et al., Bioinformatics, 2013, Hughes ME et al., PLoS Genet., 2009). A basic and recurring question for the researcher trying to balance dollars vs. detection power, is, what specifically do I gain by increasing temporal resolution and/or replicates? This should be addressed directly in the manuscript text and/or by package vignette.
● The authors “plan to extend Simphony to simulate nonstationary trends (e.g., damped rhythms) and to accommodate different periods for different genes within a simulation.” This would make a strong addition to THIS initial publication!

Annotated reviews are not available for download in order to protect the identity of reviewers who chose to remain anonymous.

---

## Round 0.2 · Minor Revisions

The reviewers have made some additional comments about your manuscript but I do not expect that these will be too difficult for you to address and I look forward to seeing your revised manuscript with a point-by-point response to each of their comments.

As you can see, Reviewer #3 is happy with the manuscript as it stands.

Reviewer #1 raises a query about the wording “lightweight”. I think you are using this term in its software engineering context, but you may well wish to take their advice and choose a term that could not be misunderstood in other contexts, where it might be seen as pejorative. I think their other follow-up question around multiple cycles is important, but I also take what I understand to be your point that under the assumption of stationarity, a single cycle can still be used to estimate the parameters (albeit with lower precision) and detect rhythmicity (albeit with lower power). I think that this highlights the tension between analysing real-world data, where stationarity cannot, as a rule, be assumed, versus analysing simulated data to demonstrate that the simulation process operated as expected. I wonder if this point could be made clearer in the manuscript. Their other comment about the extent of this manuscript is also valid, but I think I can see your argument about other planned work. Given the different audiences, the audience for this manuscript already being well-informed and looking for a tool to achieve a particular goal, I feel that the present manuscript has a distinct purpose that can be achieved despite its unusually short length.

Reviewer #2 also raises important points, here including the question of short- and long-scale series. I think that this question overlaps with that from Reviewer #1 above and that similar statements explaining that if Simphony can work for short time series, then, in the absence of dampening or drift (or other non-stationarity), by its nature, it can also work for time series of arbitrary length would be a useful addition, perhaps around Line 89. Alternatively, you could make the point that the precision of parameter recovery and power for rhythmicity detection would improve with greater numbers of cycles and show this point empirically. Their second comment suggests to me that you still need to make it clearer that the scope of this manuscript is in the generation of data, not the detection of rhythmicity. An additional Symphony parameter for missing (presumably completely at random) data would be an option, which could be specified overall or by condition, but this could also be implemented by the user in base R without any difficulty so I will leave that possibility up to you. I agree that the addition of the R code, including showing how to write additional rhyFunc() definitions, would be helpful for the reader to be able to see here in a supplement.

Reviewer 1 ·

Basic reporting

The text refers to CircaInSilico as “lightweight”, which seems quite dismissive and harsh. Is that intended?

Experimental design

I appreciate that the figures were altered to show 2 cycles. However, I strongly disagree with the authors’ assertion that rhythmicity and period can be detected from a single cycle. There is no way to tell whether the pattern will repeat the next day without actually collecting the data. This computational tool should reflect that fact by only generating simulations that are at least 2 cycles long, and the text should make clear this important condition of requiring at least 2 cycles.

I don’t understand the authors’ statement that simulating one cycle or multiple cycles is the same. The noise can vary substantially from cycle to cycle, and it’s important to see that variation when generating simulations for testing. The presence of noise is what often complicates the detection of rhythms, making each cycle different.

Cosinor can’t fit to 1 cycle of data. It may generate some numbers, but they won’t be reliable.

Validity of the findings

The manuscript still seems really thin. I expect most readers won’t understand the fundamental purpose of this package without a more substantial example demonstrating how to use it in experimental design.

Reviewer 2 ·

Basic reporting

Manuscript has notably improved after review. However, there are still some flaws on it.

First, authors state that Simphony is a tool to simulate large-scale rhythmic data. But the paper focuses on very simple examples for shor time series. I strongly recommend to the authors to extend it (Figures) to large-scale data and compare results with those illustrated in the paper for short time series. Simphony accuracy should be assesed for those examples too. As far as I know, JTK does not detect correctly asymmetric patterns. How does it performs for sawtooth shapes? Does it work for slightly sawtooth shapes? Are you methodology ready for non-equal spaced or misssing data? On the other hand, mathematical functions used in R to simulate the examples illustrated in the paper should be included, at least in the Supplemental material, it may help to non familiar readers. Finally, I am not conviced of the novelty of this work, what are the advantages of this paper with respect to Deckard A, et al. (2013) (https://academic.oup.com/bioinformatics/article/29/24/3174/193410). It should be discussed in the paper.

Experimental design

See 1

Validity of the findings

See 1

Reviewer 3 ·

Basic reporting

The authors did a great job responding/clarifying to all my previous critiques, and I am now comfortable with publication of this manuscript.

Experimental design

Fine.

Validity of the findings

Fine.

Additional comments

The authors did a great job responding/clarifying to all my previous critiques, and I am now comfortable with publication of this manuscript.

---

## Author Rebuttal · Round 0.2

# Reviewer 1

There is a serious flaw in the design in that the length of the simulations is only 24h, one cycle, in length. Rhythmicity testing requires 2 cycles at the very least.

All the time series depicted in the main manuscript and in the supplement are all only 24 hours in length (the vignettes use only 24h as well). How can rhythmicity be tested with only one cycle? Just because a time series goes up and down over a day doesn't mean it will do it again the next day. I've had experimental results where the first day looked like the first cycle of a rhythm, but then the next day was quite different, showing it was a transitory event and not a true rhythm (with a similar pattern across all samples, so multiple samples per time point doesn't alleviate the problem). All examples should show at least 2 days.

See https://www.ncbi.nlm.nih.gov/pubmed/29098954 for guidelines and best practices, which at least one of the manuscript authors signed onto and is referenced in the text. This article states, "By definition, biological rhythms repeat. We therefore recommend collecting at least 2 complete cycles of data when detecting rhythmicity (i.e., 48 h for collections under constant conditions). The guiding principle behind this recommendation is that when identifying a rhythmic process, one would like to observe both the peak and trough repeat at least once. Simulations show that collecting fewer than 2 cycles in a time series makes the resulting data sensitive to outliers and can dramatically increase the number of false-negatives (see the "Synthetic Data for Benchmarking" section)."

For purposes of phase, waveform, and amplitude estimation, it's true that under LD entrained conditions 24h of data can be used. But rhythmicity cannot be correctly tested with only 1 day of data. How is limma detecting rhythmicity with only 1 day of data? Is it really testing for a sinusoidal waveform?

Thank you for making this point. To prevent any misunderstandings, we have revised our figures to show two complete cycles. We have also added the ability to simulate non-stationary trends such as damping.

Our original simulations and figures did not reflect a judgment against collecting at least two cycles in experimental scenarios, which we completely agree is an important guideline. However, because we had explicitly designed the simulations to have no non-stationary trends, there was absolutely no difference between simulating one cycle or multiple cycles. Showing the data as a single cycle was only a matter of convenience.

The standard cosinor model detects rhythmicity by estimating coefficients for sin(t) and cos(t). It has no problem accurately fitting a sinusoid to one cycle's worth of data (assuming there are no non-stationary trends).

In addition, in trying out the package, I couldn't find a way to simulate time series longer than 24h. *It's really important that there be an easy option for specifying length of the time series simulations.* Note that CircaInSilico does provide this feature. For use in studies after constant conditions or for LD conditions in which entrainment may not be stable (short or long T-cycles, skeleton PP, jet lag experiments, etc), multiple cycles will be very important.

We have now made it easier to simulate data from multiple cycles, and have updated the documentation accordingly. It was previously possible to simulate time series longer than the period by setting timepointsType to "specified", then specifying the exact timepoints.

Due to the serious flaw of only one cycle being simulated, the rhythmicity results shown in Figure 2 are not valid.

We have revised the analysis for Figure 2 to include data from two cycles. We also now detect rhythmicity using JTK_CYCLE instead of using limma-cosinor.

For clarity, please state explicitly in lines 94-95 that the count-dispersion relation is being used to generate the function g_ik; this direct link will help the reader make the connection.

As suggested, we now explicitly state that the mean-dispersion relation corresponds to $g$.

What choices does the user have: # days, time step, noise type and parameters, signal amplitude, phase, baseline, waveform, others? Perhaps include a table listing the options available, as a convenient summary showing the flexibility of the package.

Thank you for the suggestion. We have summarized Simphony's options in a table and have added examples to the documentation.

FigS2: just an example to illustrate options? Two simulated series for each condition and gene? Can it accurately detect the amplitudes and phases? The purported goal of the package is to provide large simulated datasets to guide experimental design, so do some examples that mimic large-scale experiments. For instance, demonstrate how to use the simulated data sets to determine how many samples are needed per time point given a certain sampling rate like every 4 hours over 2 days to achieve a desired false discovery rate.

Figure S2 shows that we can use Simphony to simulate differences in amplitude and phase (and now period) between conditions. This is an important capability, given the growing amount of circadian omics data collected from multiple conditions. Importantly, Simphony does not detect any amplitudes and phases, it only simulates the data.

We have now added results from a larger simulation, in which genes had a range of values for amplitude and phase. Our goal in this manuscript is to describe the technical details and design

choices underlying Simphony. We are working on a separate study that uses Simphony to inform experimental design and to evaluate methods for rhythm detection.

The current text seems a bit thin. Demonstrating its use by applying metacycle to a large simulated data set generated by symphony would nice to see, rather than only limma for the single validation example. Using a package like metacycle will provide an alternative rhythmicity test(s) as well as phase and amplitude estimates and could further expand the scenario in Figure S2 into an interesting test case (like determining what sampling rate with 2 days of data would be required for reliable estimation of phase).

Thank you for the suggestion. We will be dedicating larger-scale applications of Simphony to a separate manuscript. We have chosen to focus the current manuscript on Simphony itself, and have therefore kept it light intentionally.

For installing simphony, note that you can also simply type drat::addRepo('hugheylab')
Into the Console window. I couldn't get the .Rprofile approach to work, but simply pasting that line into the Console worked fine.

Thanks for the feedback. We now mention this alternative on the simphony GitHub page.

## Reviewer 2

This paper uses a suitable language and the structure and goals are clearly exposed. However, some statements require additional references and other times they are too old, see the comments made on lines 24,28 of the .pdf file. Literature review presents several gaps, see comments on the lines 34, 44 of the .pdf file.

Thank you for mentioning those papers. We have revised the manuscript accordingly.

The scientific question proposed in this paper, i.e. the simulation of circadian rhythmic genes imitating real data bases is relevant for biologists. However, this question has been already dealt in literature (see the comments on line 34 in the .pdf file); the novelty of the simulation model-based is not clear (see comments on line 60 in the .pdf file); and the procedure is only validated for symmetric shapes, despite that circadian data bases also include non-sinusoidal patterns (see line 143 in the .pdf file). Please, see Experimental Design for details.

The novelty is not in the shape of the simulated data, but in the offering of a general tool to reproducibly simulate rhythmic data. We have revised the manuscript to clarify that Simphony can handle any shape of rhythmic function that the user provides, and we have added more examples of rhythmic time-courses.

This question has been previously studied in literature, both for microarray (Dembelé (2013) among others) and RNA-seq technologies (Zappia, Phipson and Oshlack (2017) among others). Authors do not refer/compare to this fact in the paper.

Those packages are designed to simulate microarray or scRNA-seq data, whereas simphony is a specialized tool that makes it much easier to simulate rhythmic data from time-course experiments. We have clarified this in the Introduction.

The model employed to generate synthetic data is very close to Cosinor model, at least for the default case considered in this paper, i.e. when f is assumed to be a sinusoidal function. Authors should discuss the similarities/differences with this model. In addition, only this default (sinusoidal) function is considered to validate the procedure. It is not enough to replicate the wide variety of patterns that appear in practice.

The default rhythmic function in Simphony is a sine wave, which corresponds exactly to the shape of rhythmicity fit by the cosinor model. Thus, the cosinor model is the appropriate choice to determine whether the data generated by Simphony corresponds to the parameters it is given. It is outside the scope of this manuscript to reproduce the spectrum of rhythmic patterns that have been observed in experimental data.

As expected, results show a suitable performance of the simulation tool. Note that data were generated from a sinusoidal model and the fittings are made with Cosinor model (based on sinusoidal shape too). However, non-sinusoidal (asymmetric) rhythms are found in circadian data bases (Thaben 2014). Although, authors consider a sawthooth wave, it is again just a symmetric pattern. More arbitrary and extreme patterns are required to validate the procedure.

We have added more examples of rhythmic patterns to the manuscript. We have thoroughly validated Simphony, but we are not attempting to exhaustively evaluate rhythm detection. We have clarified this point in the manuscript. The sawtooth wave is not symmetric; it takes the entire period to go up, but takes no time to go down.

The originality of the research question is questionable.
Conclusions depends on the periodic simulation function (which is always symmetric) and on the Cosinor regression model which intrinsically assumes sinusoidal shapes. Thus, those conclusion cannot be extended to circadian data bases where there are many arbitrary rhythmic patterns, not only sinusoidal.

The goal of this manuscript is to describe Simphony the simulation package and to give basic examples of its use. We are not attempting to evaluate the ability of particular methods to detect rhythms of various shapes, and are thus not making conclusions about rhythm detection. We have revised the manuscript accordingly.

Line 24: There are many other recent references. For example: Zhang et al (2014) or Cornelissen and Otsuka (2017)

We have added the Zhang reference as a more recent example of the ubiquity of circadian systems. The Cornelissen and Otsuka paper reviews aging trends of rhythms rather than their widespread distribution, so we have not cited it.

Line 26: Vague statement. What technologies, do you refer to microarray, scRNA-Seq? Please, mention at least those that you are going to focus on the paper, as well as the molecules that are going to mesure in each case

We have added transcriptome analysis using RNA-Seq as an example. However, we are not measuring any molecules, and most of Simphony is agnostic with respect to the technology behind the experimental data being simulated.

Line 28: There are numerous, but you only provide one reference

We have now cited additional rhythm detection methods.

Line 33: This reference is not directly related to the statement. Please, replace or give a better explanation

We have replaced this reference and clarified our statement.

Line 34: Please, include some references for this statement. I do not totally agree. There are simulation studies in literature imitating realistic features which are computationally demanding, see Dembélé, 2013 (for microarrays) or Zappia, Phipson and Oshlack (2017) for (scRNA-Seq).

We have moderated this statement to say that simulations are *typically* faster and less expensive. Given the time and costs associated with wet-lab experiments (e.g., culturing cells, caring for mice, library prep, sequencing) compared to running a computer program, we feel this statement is reasonable without a specific reference.

Line 44: Do you know about iasva R-package (https://github.com/dleelab/iasvaExamples)? This methodology improves Polyester

We have read the documentation for Iteratively Adjusted Surrogate Variable Analysis (IA-SVA). IA-SVA inputs parameters derived from human pancreatic samples into Polyester for simulating single cell data. It is not a generalizable improvement to Polyester's methodology.

Line 49: Can you ensure that it replicates arbitrary rhythmic patterns in data bases (microarray or RNA-Seq) I did not see any evidence in the paper

We have given Simphony the flexibility to simulate any shape of stationary or non-stationary rhythm. It is not the goal of the present manuscript, however, to replicate the diversity of rhythmic patterns observed in experimental transcriptome data. We have clarified this point in the Conclusions.

Line 60: This model used for abundance resembles  Cosinor model, isn't it? What are the differences???. Can your model be reduced to Cosinor, why not?

The model is only equivalent to cosinor if the rhythmic function is a sinusoid. We have clarified this point.

Line 61: Please, what are the other periodic functions you have considered? In practice, there are a wide variety of rhythmic patterns.

We now show simulations based on additional rhythmic functions. Importantly, Simphony can simulate rhythms according to any pattern that can be encoded as a function in R.

Line 66: Do you refer to (sc)RNA-Seq? Please, be more precise

We have clarified this to "RNA-seq." We are not referring specifically to single-cell RNA-seq.

Line 69: Have you tried with different values, isn't it?

Yes, we have validated the simulated data for various values of the standard deviation, as shown in Table S1.

Line 83: I consider that these detials may be included in the supplementary and this part can be reduced to the basics of the procedure. Reader can be confussed

Those details are part of the Materials and Methods section, and we would prefer not to separate them.

Line 113: Time sampling and periodic function (f) should be also taken into account to validate the procedure

We now show simulations based on additional rhythmic functions. The Simphony R package includes an extensive suite of unit tests to validate that our time sampling is correct.

Line 115: You simulate sinusoidal   patterns and then you fit genes using Cosinor, a sinusoidal-based model. So it is expected to work well. But, What happen if data are generated according to a periodic but non-symmetric function?

We have revised this analysis to detect rhythmicity using JTK_CYCLE, not limma-cosinor. We are not attempting to comprehensively evaluate non-sinusoidal rhythm detection, we only wish to provide a simple example of how one can use simulated data from Simphony.

Line 140: The sawtooth wave that you illustrate is very symmetric. Please, replace it or add more asymmetric shapes

We have now added simulations that have other rhythmic shapes. The sawtooth wave is not symmetric; it takes the entire period to go up, but takes no time to go down.

Line 143: In omics experiments, there are evidences that rhythmic features not only display sinusoidal patterns, see among others Thaben 2014.
Thus, Cosinor model may not be enough for those cases.

We have revised this analysis to detect rhythmicity using JTK_CYCLE, not limma-cosinor. It is outside the scope of this manuscript to comprehensively evaluate non-sinusoidal rhythm detection.

# Reviewer 3

Simphony generates two rhythmic waveforms. Are more necessary? In 2013, Deckard A. et al. reported a simulation generating six different periodic patterns and two non-periodic patterns.

Simphony is not limited to two rhythmic waveforms. Although we only showed two examples in the original manuscript, Simphony can simulate rhythmicity according to any waveform that the user desires (and that can be encoded as a function in R). We have revised the Results section to make this more clear and have added examples that have non-stationary trends.

While free to download and use in R stats, Simphony is still not as user-friendly as CircaInSilico (Hughes et al, JBR 2017), which can be run as a web-based App (no coding required). This could ease the learning curve and encourage wider adoption.

We concede that an R package, although standard in bioinformatics, is not as friendly to non-coders as a web app. However, we would argue that most non-coders don't want to simulate data as much as they want to know what the simulations would tell them. Therefore, as part of our upcoming and separate manuscript, we plan to make a web app to explore our larger evaluation of experimental design and rhythm detection. The simphony package and this manuscript will stand on their own.

The manuscript lacks explicit description of how Simphony could be used to help guide the design of a typical time-course experiment. The ROC curves (Fig 2) demonstrate nicely how cosinor regression better discriminates rhythmic from non-rhythmic features when the sampling interval is smaller (and amplitudes are higher). This, however, is pretty intuitive, and a

conclusion reached in prior work (e.g. Deckard et al., Bioinformatics, 2013, Hughes ME et al., PLoS Genet., 2009). A basic and recurring question for the researcher trying to balance dollars vs. detection power, is, what specifically do I gain by increasing temporal resolution and/or replicates? This should be addressed directly in the manuscript text and/or by package vignette.

Indeed, we used this as an example precisely because the result is intuitive and consistent with prior work (although we have now switched this analysis to use JTK instead of limma-cosinor). The focus of this manuscript is Simphony itself; we plan to address experimental design in a separate manuscript.

The authors "plan to extend Simphony to simulate nonstationary trends (e.g., damped rhythms) and to accommodate different periods for different genes within a simulation." This would make a strong addition to THIS initial publication!

Thank you for this encouragement. We have now added these features to Simphony and have revised the manuscript accordingly.

---

## Round 0.3 · accepted · Accept

Your revisions have addressed all of the outstanding comments. I look forward to seeing any new features you decide to add to Simphony in the future.

#